# Optimised Non-Coding Regions of mRNA SARS-CoV-2 Vaccine CV2CoV Improves Homologous and Heterologous Neutralising Antibody Responses

**DOI:** 10.3390/vaccines10081251

**Published:** 2022-08-04

**Authors:** Nicole Roth, Jacob Schön, Donata Hoffmann, Moritz Thran, Andreas Thess, Stefan O. Mueller, Benjamin Petsch, Susanne Rauch

**Affiliations:** 1CureVac AG, 72076 Tübingen, Germany; 2Institute of Diagnostic Virology, Friedrich-Loeffler-Institut, 17493 Greifswald-Insel Riems, Germany

**Keywords:** SARs-CoV-2, mRNA vaccine, COVID-19, COVID-19 variant, immunogenicity, virus neutralising antibody titre

## Abstract

More than two years after the emergence of SARS-CoV-2, 33 COVID-19 vaccines, based on different platforms, have been approved in 197 countries. Novel variants that are less efficiently neutralised by antibodies raised against ancestral SARS-CoV-2 are circulating, highlighting the need to adapt vaccination strategies. Here, we compare the immunogenicity of a first-generation mRNA vaccine candidate, CVnCoV, with a second-generation mRNA vaccine candidate, CV2CoV, in rats. Higher levels of spike (S) protein expression were observed in cell culture with the CV2CoV mRNA than with the CVnCoV mRNA. Vaccination with CV2CoV also induced higher titres of virus neutralising antibodies with accelerated kinetics in rats compared with CVnCoV. Significant cross-neutralisation of the SARS-CoV-2 variants, Alpha (B.1.1.7), Beta (B.1.351), and the ‘mink’ variant (B1.1.298) that were circulating at the time in early 2021 were also demonstrated. In addition, CV2CoV induced higher levels of antibodies at lower doses than CVnCoV, suggesting that dose-sparing could be possible with the next-generation SARS-CoV-2 vaccine, which could improve worldwide vaccine supply.

## 1. Introduction

The ongoing COVID-19 pandemic has created a global need for the fast development of vaccines. By 4 February 2022, 33 vaccines against SARS-CoV-2 have received authorisation in 197 countries, and 141 vaccine candidates are currently in clinical development [1,2]. However, the emergence of variants of concern (VOCs) of SARS-CoV-2, potentially able to escape pre-existing immunity, highlights the need for the continuous improvement and adaptation of vaccines [3,4,5,6,7,8,9]. The mRNA vaccine platform is a promising technology to meet this need, since it is a highly versatile, adaptable technology that has been shown to provide efficacious and effective vaccines against SARS-CoV-2 [10,11,12,13].

CVnCoV and one of the next-generation vaccine candidates developed by CureVac, CV2CoV, are sequence optimised, capped, chemically unmodified mRNA-based vaccines based on CureVac’s mRNA technology platform, RNActive^®^ [14,15,16,17], and encapsulated using identical lipid nanoparticle (LNP) technology. Both encode full-length SARS-CoV-2 spike (S) proteins with pre-fusion stabilising K986P and V987P mutations [18,19]. The vaccines differ in their non-coding elements flanking the open reading frame (ORF), i.e., the 5′ and 3′ untranslated regions (UTRs) and the 3′ tail. These changes were introduced to improve intracellular stability and translation of the mRNA, and thereby increase and prolong protein expression and thus improve immunogenicity. Other vaccines based on CureVac’s RNActive^®^ technology have been shown to provide high protein expression and efficacious adaptive immune responses against different infectious diseases in both preclinical and clinical studies [20,21,22]. Preclinical studies in non-human primates (NHPs) demonstrated improved immunogenicity and protective efficacy of CV2CoV compared with CVnCoV [23]. Here, we report the levels of SARS-CoV-2 S protein expression produced by CVnCoV and CV2CoV mRNA in cell culture and the humoral responses induced by both in LNP-formulations in an outbred rat model. 

## 2. Materials and Methods

### 2.1. mRNA Vaccines

Both CVnCoV and CV2CoV mRNAs, produced using the RNActive^®^ platform, have a 5′ cap1 structure, GC-enriched open reading frame (ORF), 3′ UTR, a polyA stretch, and chemically unmodified nucleosides [14]. CVnCoV mRNA contains part of the 3′ UTR of the human alpha haemoglobin gene as 3′ UTR followed by a polyA stretch, a C30 stretch, and a histone stem loop. CV2CoV contains a 5′ UTR from the human hydroxysteroid 17-beta dehydrogenase 4 gene (HSD17B4) and a 3′ UTR from the human proteasome 20S subunit beta 3 gene (PSMB3), followed by a histone stem loop and a polyA stretch. The mRNA was encapsulated with lipid nanoparticle (LNP) technology from Acuitas Therapeutics (Vancouver, BC, Canada). The LNPs used in these vaccines were particles of ionisable amino lipid, phospholipid, cholesterol, and a PEGylated lipid. Both mRNAs encoded the full-length spike glycoprotein of SARS-CoV-2 featuring pre-fusion stabilising K_986_P and V_987_P mutations (NCBI Reference Sequence NC_045512.2, GenBank accession number YP_009724390.1).

### 2.2. In Vitro Protein Expression

HeLa cells were seeded in 6-well plates at a density of 400,000 cells/well and 24 h later were transfected with 2 µg of CVnCoV or CV2CoV mRNA per well using lipofection. For this, the mRNAs were complexed with Lipofectamine 2000 (Life Technologies, Carlsbad, CA, USA) at a ratio of 1:1.5 and transfected into cells according to the manufacturer’s protocol. Protein expression was assessed 24 h post-transfection. The cells were fixed and analysed with intact (surface staining) or permeabilised plasma membranes by treatment with Perm/Wash buffer (BD Biosciences, Cat. 554723) (intracellular staining) for fluorescence activated cell sorting (FACS) analysis. Specific S protein expression was assessed after staining with human anti-SARS-CoV-2 S antibody (CR3022) (Creative Biolabs, Shirley, NY, USA, Cat. MRO-1214LC) followed by goat anti-human IgG F(ab’)_2_ fragment PE antibody (Immuno Research, Mendota Heights, IL, USA, Cat. 109-116-097) in a BD FACSCanto II cell analyser and FlowJo 10 software.

### 2.3. Animals

Female and male outbred Wistar rats, aged 7–8 weeks, were provided and handled by Preclinics Gesellschaft für präklinische Forschung mbH, (Potsdam, Germany). The animal study was conducted in accordance with German laws and guidelines for animal welfare and the protocol received the appropriate local and national ethics approvals (2347-14-2018 LAVG Brandenburg). 

### 2.4. Vaccination 

The Wistar rats received two doses (day (D)0, D21) of CV2CoV, CVnCoV, or NaCl as a negative control, injected into the gastrocnemius muscle. Five mRNA concentrations, ranging from 0.5 to 40 µg of CV2CoV or CVnCoV, were assessed.

### 2.5. Antibody Analysis

Blood samples were taken on D0 before the first dose, on D14, on D21 (before the second dose), and on D41. SARS-CoV-2 spike receptor-binding domain (RBD) protein-specific IgG1 and IgG2a binding antibodies were detected using ELISA. The plates were coated with 1 µg/mL of SARS-CoV-2 (2019-nCoV) spike RBD-His (Sino Biologicals, Beijing, China, Cat. 40592-V08B) for 4–5 h at 37 °C, and then blocked overnight in 1% BSA, washed and incubated with serum for 2 h at room temperature. After washing, biotinylated mouse anti-rat IgG1 (Biolegend, San Diego, CA, USA, Cat. 407404) or biotinylated mouse anti-rat IgG2a (Biolegend, Cat. 407504) was added to the wells and the plate incubated for 1 h at room temperature, followed by washing and incubation for 30 min at room temperature with horseradish peroxidase (HRP)-streptavidin (BD, Franklin Lakes, NJ, USA, Cat. 554066) and Amplex red substrate for 45 min at room temperature. The detection was performed in a BioTek Synergy HTX plate reader, with a 530/25-excitation filter, a 590/35-emission filter, and a sensitivity setting of 45. 

At VisMederi srl (Siena, Italy), virus neutralising antibody (nAb) titres for ancestral SARS-CoV-2 were determined in heat-inactivated sera (56 °C for 30 min) tested in duplicate at a starting dilution of 1:10 followed by serial two-fold dilutions. The diluted sera were incubated with 100 median tissue culture infectious doses (TCID_50_) of ancestral SARS-CoV-2 (strain 2019-nCov/Italy-INMI1 (GISAID accession EPI_ISL_410545) clade O, with no amino acid substitution for D614) for 1 h at 37 °C. Every 96-well plate had eight wells containing only cells and medium as the cell control and eight wells containing only cells and virus as the virus control. Infectious virus was quantified by incubation of 100 µL of the virus-serum mixtures on a confluent layer of Vero E6 cells (ATCC, Manassas, VA, USA, Cat. 1586) followed by incubation for 3 days at 37 °C. The cytopathogenic effect (CPE) was then scored under a microscope. A back titration was performed for each run to verify the correct TCID_50_ range of the working virus solution. Virus nAb titres were calculated according to the method described by Reed & Muench [24]. An arbitrary value of 5 was attributed if no neutralisation was observed (MNt < 10). Virus nAb titres were expressed as the neutralising dose (ND) 100 of the sera.

At the Friedrich-Loeffler-Institut (Greifswald-Insel Riems, Germany), Virus nAb titres for ancestral SARS-CoV-2 and variant strains were determined in heat-inactivated sera (56 °C for 30 min) pre-diluted 1:16 with DMEM. Three 100 µL aliquots were transferred to a 96 well plate. Serial two-fold dilutions were performed in the plate by mixing 50 µL of the serum dilution with 50 µL DMEM. Subsequently, 50 µL of the dilutions (100 TCID_50_/well) of ancestral SARS-CoV-2 (SARS-CoV-2 Germany/BavPat1/2020 (GISAID accession EPI_ISL_406862) clade G, with amino acid substitution at position 614, D614G) or the variants, B.1.1.298, [25] (SARS-CoV-2 hCoV19/DK/Cl-5/1; GISAID accession EPI_ISL_616802 ref) originating from Denmark, B.1.1.7 (Alpha; SARS-CoV-2 hCoV-19/Germany/NW-RKI-I-0026/2020 GISAID accession EPI_ISL_803957) originating from the UK, and B.1.351 (Beta; SARS-CoV-2 hCoV-19/Germany/NW-RKI-I-0029/2020 (GISAID accession EPI_ISL_803957)) originating from South Africa was added to each well and incubated for 1 h at 37 °C. Lastly, 100 µL of trypsinated Vero E6 cells (cells from one confluent TC175 flask in 100 mL) in DMEM with a 2% penicillin/streptomycin supplementation was added to each well. After incubation for 72 h at 37 °C, the wells were evaluated using light microscopy. A serum dilution was counted as the neutralising antibody (nAb) titre in the absence of a visible CPE. The virus titre was confirmed using virus back titration, with positive and negative serum samples included. 

## 3. Results

### 3.1. In Vitro Protein Expression

The expression of intracellular and cell surface S protein was 3.3- and 1.8-fold higher, respectively, in HeLa cells transfected with CV2CoV mRNA than in those transfected with the same amount of CVnCoV mRNA (Figure 1).

### 3.2. Safety and Toxicity

No safety or toxicity concerns were observed in the rats used in this study. In particular, no differences in body weight were detected in the rats who received CVnCoV vs. CV2CoV.

### 3.3. In Vivo Immune Response in Wistar Rats

The antibody response directed against the S receptor-binding domain (RBD) developed at lower doses of CV2CoV and was generally statistically significantly higher after CV2CoV vaccination compared with CVnCoV. A dose-dependent RBD antibody response was observed for doses of 0.5 µg, 2 µg, and 8 µg of CV2CoV (Figure 2). Higher levels of antibodies against RBD were observed with 20 µg and 40 µg of CV2CoV, without a clear dose-response. Antibodies against RBD developed rapidly and were detectable two weeks after a single injection in all dose groups in CV2CoV-vaccinated rats and in the 20 µg and 40 µg groups in CVnCoV-vaccinated rats (Figure 2A). A clear boost effect was observed on D42 after the second dose of both CVnCoV and CV2CoV (Figure 2C).

The induction of virus nAb titres in rat sera was assessed in a cytopathic effect (CPE)-based assay with homologous ancestral SARS-CoV-2 (Figure 3). CV2CoV, but not CVnCoV, induced significant, dose-dependent neutralising titres two weeks after a single dose in all animals vaccinated with a dose of 2 µg or higher (Figure 3A). On D14, CV2CoV induced homologous titres ranging from 1:42 in the 2-µg group to 1:193 in the 40-µg group. The second dose led to substantial increases in virus nAb titres in all CV2CoV vaccinated groups except for two animals in the 0.5 µg group. Two doses of CVnCoV also induced detectable virus nAb titres although the levels were significantly lower and more dispersed within the groups than with CV2CoV (Figure 3C). On D42, the virus nAb titres in animals that had received ≥8 µg of CV2CoV exceeded the upper range of detection in the assay, i.e., a dilution of 1:5120 (Figure 3C).

CV2CoV doses of >2 µg induced higher virus nAb titres than 20 µg of CVnCoV against ancestral SARS-CoV-2 and the variants B1.1.298, B1.1.7, and B1.351 on D42 (Figure 4). For example, the median titres induced in the CV2CoV 8 µg dose group were 1:2438 for ancestral SARS-CoV-2 (Figure 4A), 1:4096 for B.1.1.298 (Figure 4B), 1:1935 for B1.1.7 (Figure 4C), and 1:806 for B.1.351 (Figure 4D) compared with 1:161, 1:406, 1:256 and 1:64, respectively, in the CVnCoV 20 µg group.

## 4. Discussion

The higher protein expression in cell culture, observed with CV2CoV mRNA compared with CVnCoV mRNA, was consistent with the higher levels of RBD antibodies and nAb titres induced by the CV2CoV vaccination in the outbred Wistar rats. This emphasises the utility of results from in vitro cell-based protein expression studies as a criterion for mRNA candidate selection. Doses as low as 0.5 µg of CV2CoV induced detectable antibody responses, and doses of ≥2 µg resulted in higher antibody levels than those observed with up to 20 µg of CVnCoV. Virus nAb titres were detectable as early as two weeks after a single dose of ≥2 µg of CV2CoV, whereas two doses of CVnCoV were required to induce detectable levels of virus nAb titres in this study and in previous studies in mice, hamsters and NHPs, demonstrating the enhanced immunogenicity of CV2CoV [23,26,27]. Comparing different animal models, we clearly see that CV2CoV induced faster and higher titres of binding and virus neutralising antibodies, compared with CVnCoV, independent of the animal model used. The main advantage of using a rat model is that larger volumes of sera can be obtained, and this is increasingly important as there is a need to test neutralising activity against an increasing panel of SARS-CoV-2 isolates as more variants emerge.

Compared with CvnCoV, CV2CoV induced a stronger cross-neutralising response against three SARS-CoV-2 variants, B.1.1.298, B.1.1.7, and B.1.351 that were circulating at the time these experiments were performed in early 2021. The nAb titres induced by CV2CoV for ancestral SARS-CoV-2, B.1.1.7, and B.1.1.298 were comparable, but those for B.1.351 were lower, in line with previous observations [28]. These results are consistent with those observed in NHPs vaccinated with CVnCoV or CV2CoV that also showed higher cross-neutralising titres for several variants, including the B.1.617.2 (Delta) variant in NHPs vaccinated with CV2CoV [23]. The levels of cross-neutralisation titres values are comparable to, or better than, those previously reported [29,30,31].

In summary, CV2CoV induced neutralising antibodies in rats more rapidly and at lower doses than CVnCoV, i.e., within two weeks after the first dose. These results suggest that dose-sparing may be possible with the next-generation SARS-CoV-2 vaccines, which could contribute to improving worldwide vaccine supplies. Moreover, the efficacy of CV2CoV at lower doses could also be an advantage for the future development of multivalent and combination vaccines, in which each valency would need to be efficacious at a lower dose. Overall, CV2CoV and other next-generation SARS-CoV-2 vaccines represent potential candidates for further clinical development. The improved expression and immunogenicity of these next-generation SARS-CoV-2 vaccines will be important tools for tackling the SARS-CoV-2 pandemic. 

## Figures and Tables

**Figure 1 vaccines-10-01251-f001:**
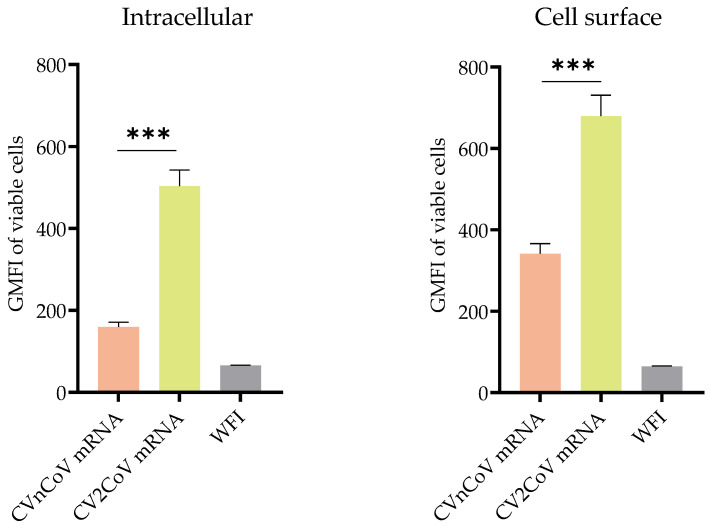
Comparison of protein expression induced by CVnCoV and CV2CoV mRNA. Expression of SARS-CoV-2 spike (S) protein in HeLa cells transfected with CVnCoV or CV2CoV mRNA, as indicated, detected using flow cytometric analysis via an S-specific antibody either with or without membrane permeabilisation, allowing detection of intracellular and cell surface expression, respectively. The geometric mean fluorescence intensity (GMFI) is expressed as mean + standard deviation (SD) of duplicate samples from two independent experiments. *** = *p* < 0.001 (*t*-test). WFI: water for injection.

**Figure 2 vaccines-10-01251-f002:**
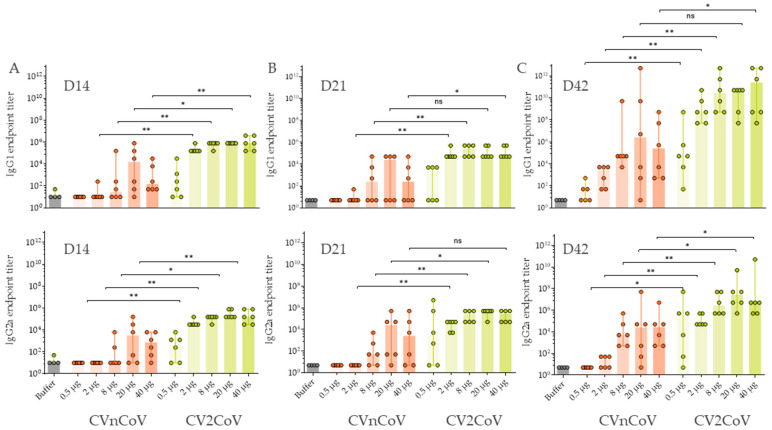
CV2CoV induced higher levels of antibodies against the S receptor-binding domain (SRBD) than CVnCoV in rats. Groups of female and male Wistar rats (*n* = 6/group) were vaccinated IM on day (D)0 and D21 with five different doses ranging from 0.5 µg to 40 µg of CVnCoV or CV2CoV. Wistar rats (*n* = 4) vaccinated with 0.9% NaCl (Buffer) on D0 and D21 served as negative controls. SRBD-specific antibody responses are shown as endpoint titres for IgG1 and IgG2a in serum after one dose ((**A**) D14 and (**B**) D21) or two doses ((**C**) D42). Each dot represents an individual animal, and the vertical lines show the range. A Mann–Whitney test was used to compare the groups: ns = >0.5, * = *p* < 0.05, ** = *p* < 0.01.

**Figure 3 vaccines-10-01251-f003:**
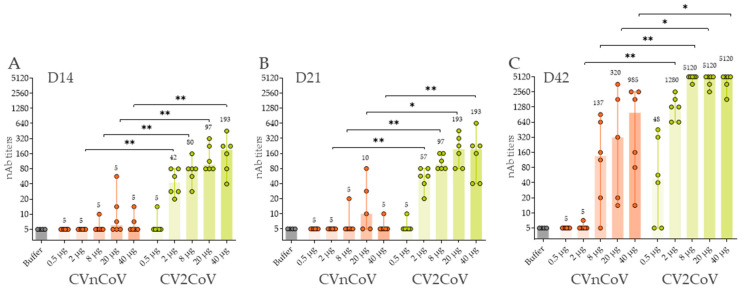
CV2CoV induced higher virus neutralising antibody (nAb) titres against ancestral SARS-CoV-2 than CVnCoV. Groups of female and male Wistar rats (*n* = 6/group) were vaccinated IM with five different doses ranging from 0.5 µg to 40 µg of CVnCoV or CV2CoV on D0 and D21. Rats (*n* = 4) vaccinated with 0.9% NaCl (buffer) served as negative controls. SARS-CoV-2 virus neutralising antibody (nAb) titres in serum samples taken on (**A**) D14, (**B**) D21 and (**C**) D42 were analysed. The highest dilution was 1:5120. Each dot represents an individual animal, and vertical lines show the range. Mann-Whitney test was used to compare groups. * = *p* < 0.05, ** = *p* < 0.01.

**Figure 4 vaccines-10-01251-f004:**
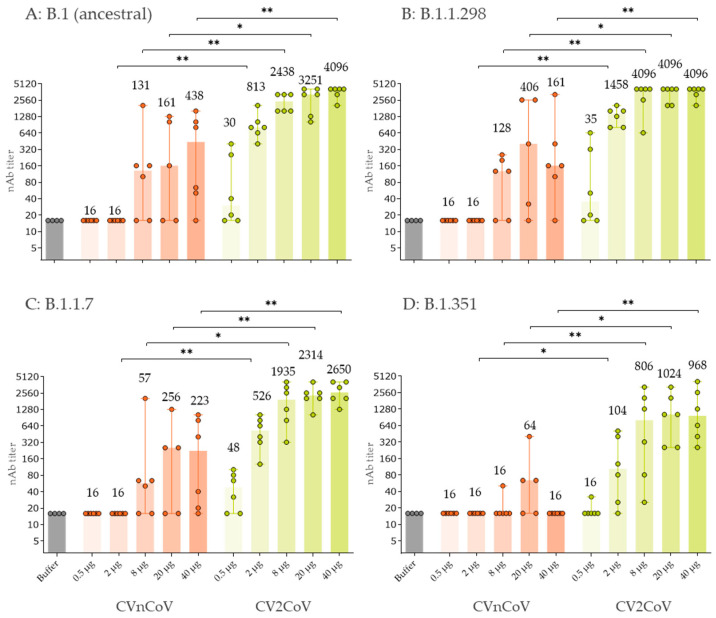
Two doses of CV2CoV on D0 and D21 induced higher virus neutralising antibody (nAb) titres on D42 against (**A**) ancestral SARS-CoV-2 and the variants (**B**) B1.1.298, (**C**) B.1.1.7 and (**D**) B1.351 compared with two doses of CVnCoV. Groups of female and male Wistar rats (*n* = 6/group) were vaccinated IM with five different doses ranging from 0.5 µg to 40 µg of CVnCoV or CV2CoV on D0 and D21. Rats (*n* = 4) vaccinated with 0.9% NaCl (Buffer) served as negative controls. SARS-CoV-2 virus neutralising antibody (nAb) titres in serum samples taken on D42 were analysed. The highest dilution was 1:5120. Each dot represents an individual animal. The median nAb titres are shown above each bar and the vertical lines show the range. Mann-Whitney test was used to compare groups. * = *p* < 0.05, ** = *p* < 0.01.

## Data Availability

The data presented in this study are available from the corresponding author on reasonable request.

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
