# Peer review of "Optimised Non-Coding Regions of mRNA SARS-CoV-2 Vaccine CV2CoV Improves Homologous and Heterologous Neutralising Antibody Responses"

_vaccines, 2022, doi:10.3390/vaccines10081251_

Round 1
Reviewer 1 Report
In this manuscript, Roth et al. compared the in vitro expression of and rat antibody response elicited by the first and second generation of CureVac’s COVID-19 vaccine. They reported the second generation of CureVac’s vaccine, of which the non-coding regions of the mRNA was modified, expressed Spike protein at higher level than the first generation of CureVac vaccine, and induced faster and stronger binding and neutralizing antibodies as well. Considering the disappointing clinical trial results of the first generation of CureVac COVID-19 vaccine, the findings presented in this manuscript suggest the 2nd generation of the CureVac vaccine warrant further development for human vaccination.
Major points:
1. To make the manuscript more relevant for current situation, it would be good if the authors can use the rat’s sera to check nAb against new VOCs, e.g., Omicron BA.1, BA.2, BA. 4 etc. Pseudovirus neutralization results will be good enough.
2. Did the authors see any safety/toxicity concerns in these experiments? E.g., rat body weight change etc. and any difference between the two generations of the vaccine in this respect?
Minor points:
1. Fig. 1, Y-axis, should be labelled as “GMFI of viable cells”.
2. Fig. 3, What ancestral virus used here, Italy or Germany strain? What clade, A or B? B.1 (D614G)? Same for Fig. 4 ancestral virus.
3. Fig. 4, should be labelled with panel A-D.
Reviewer 2 Report
The authors have already published two papers comparing extensively the efficacy of these two mRNA vaccines in human ACE2 transgenic mice in “Nature Communications” and in non-human primates in “Nature”, respectively. The main novelty of the current manuscript is probably the use of another animal model: outbred Wistar rats.
Most part of the current Discussion section is reiterating the results already in the results section. The authors should probably focus on discussing their findings. Is there any difference in the immune responses observed between the animal models?
The authors mentioned that accelerated kinetics of antibody responses was observed for their second-generation vaccine. Where is the evidence? Perhaps they can look at the antibody response even earlier than 14th day and even check the IgM antibody responses?
Please label the panels of Figure with A, B, C and D as they are referred in the body text.
Please consider replacing the “homogeneous and heterogeneous” with “homologous and heterologous” in the title.
Author Response
Reviewer 2: The authors have already published two papers comparing extensively the efficacy of these two mRNA vaccines in human ACE2 transgenic mice in “Nature Communications” and in non-human primates in “Nature”, respectively. The main novelty of the current manuscript is probably the use of another animal model: outbred Wistar rats.
Comment 2.1: Most part of the current Discussion section is reiterating the results already in the results section. The authors should probably focus on discussing their findings. Is there any difference in the immune responses observed between the animal models?
Response 2.1: We have added the following text about the different animal models: ‘Comparing the different animal models we clearly see that CV2CoV induced faster and higher titres of binding and virus neutralising antibodies, compared with CVnCoV, independent of the animal model used. The main advantage of using a rat model is the larger volumes of sera that can be obtained, and this is increasingly important as there is a need to test neutralising activity against an increasing panel of SARS-CoV-2 strains as more variants emerge.‘
Comment 2.2: The authors mentioned that accelerated kinetics of antibody responses was observed for their second-generation vaccine. Where is the evidence? Perhaps they can look at the antibody response even earlier than 14th day and even check the IgM antibody responses?
Response 2.2: We did not take blood before Day 14 in this study and we did not look at IgM responses. However, as we state in the manuscript: ‘Virus nAb titres were detectable as early as two weeks after a single dose of ≥2 µg of CV2CoV, whereas two doses of CVnCoV were required to induce detectable levels of virus nAb titres in this study and in previously studies in mice, hamsters and NHPs, demonstrating enhanced immunogenicity of CV2CoV.’ We interpreted that the absence of nAbs on Day 14 in animals receiving CVnCoV, but their present in animals receiving the CV2CoV vaccine was evidence that the response is more rapid. We also interpreted that the observation that two doses of CV2CoV induces higher nAb titres responses than two doses of CVnCoV was evidence of a stronger response of CV2CoV compared to CVnCoV.
Comment 2.3: Please label the panels of Figure with A, B, C and D as they are referred in the body text.
Response 2.3: The labels have been added to Fig 4.
Comment 2.4: Please consider replacing the “homogeneous and heterogeneous” with “homologous and heterologous” in the title.
Response 2.4: The title has been changed as suggested.
